# Particulate Matter Emissions of Four Different Cigarette Types of One Popular Brand: Influence of Tobacco Strength and Additives

**DOI:** 10.3390/ijerph16020263

**Published:** 2019-01-17

**Authors:** Markus Braun, Friedemann Koger, Doris Klingelhöfer, Ruth Müller, David A. Groneberg

**Affiliations:** 1Institute of Occupational, Social and Environmental Medicine, Goethe University Frankfurt, Theodor-Stern-Kai 7, D-60590 Frankfurt am Main, Germany; f.koger@gmx.de (F.K.); klingelh@med.uni-frankfurt.de (D.K.); ruth.mueller@med.uni-frankfurt.de (R.M.); groneberg@med.uni-frankfurt.de (D.A.G.); 2Medical Entomology, Department of Biomedical Sciences, Institute of Tropical Medicine, Nationalestraat 155, B-2000 Antwerpen, Germany

**Keywords:** second-hand smoke, environmental tobacco smoke, particulate matter, additives, cigarette strength

## Abstract

The inhalation of particulate matter (PM) in second-hand smoke (SHS) is hazardous to health of smokers and non-smokers. Tobacco strength (amount of tar, nicotine, and carbon monoxide) and different additives might have an effect on the amount of PM. This study aimed to investigate the influence of tobacco strength or additives on PM. Four cigarette types of the brand Marlboro with different strengths and with or without additives were analyzed in comparison to the 3R4F reference cigarette. SHS was generated by an automatic environmental tobacco smoke emitter (AETSE) in an enclosed space with a volume of 2.88 m³. PM concentrations (PM_10_, PM_2.5_, PM_1_) were measured with a laser aerosol spectrometer followed by statistical analysis. The two strongest Marlboro brands (Red and Red without additives) showed the highest PM concentrations of all tested cigarettes. The measured mean concentrations Cmean of PM_10_ increased up to 1458 µg/m³ for the Marlboro Red without additives (PM_2.5_: 1452 µg/m³, PM_1_: 1263 µg/m³). The similarly strong Marlboro Red showed very similar PM values. The second strongest type Marlboro Gold showed 36% (PM_10_, PM_2.5_) and 32% (PM_1_) lower values, respectively. The “lightest” type Marlboro Silver Blue showed 54% (PM_10_, PM_2.5_) or 50% (PM_1_) lower PM values. The results indicate that the lower the tar, nicotine, and carbon monoxide amounts, as well as the longer the cigarette filter, the lower are the PM levels. An influence of additives could not be determined.

## 1. Introduction

Since the beginning of the 20th century tobacco consumption increased steadily worldwide. Approximately 1.1 billion people aged 15 or older are current smokers worldwide. Meanwhile, smoking is one of the most important avoidable causes of premature death in the world. By now, more than 7 million people are killed each year owing to tobacco use, whereby 890,000 of them are non-smokers exposed to second-hand smoke (SHS) [1], also called environmental tobacco smoke.

SHS as a composite of exhaled smoke from the smoker and mostly side-stream smoke from the smoldering tobacco product [2,3] is the major risk factor for indoor air pollution [4] and one of the main causes of avoidable lung cancer [5]. SHS is also a major origin of airborne particulate matter (PM) [6]. The adverse health effects of PM, especially cardiovascular and respiratory diseases [7,8,9] increase in relation to PM exposure [10]. This applies also for human skin diseases [11], breast cancer mortality [12], and risk of ischemic stroke [13]. In addition, PM is more harmful to health of children and infants because of their smaller body weight. The dimension of the upper respiratory tract of infants is smaller than that of adults. Especially ultrafine particles (UFPs, particles < 100 nm) could be concentrated in the head region and translocate to the brain via the olfactory bulb [14].

PM as a mixture of differently sized liquid and solid particles varies in source and composition [15]. One possibility to classify PM is by the particle size and defines the deepness of the penetration in to the respiratory tract. The smaller the particles the deeper they penetrate and the more severely are the health effects [16,17]. Furthermore, smaller particles have a higher ability to adsorb toxic organic molecules and UFPs can penetrate through the blood and nervous system into the brain and diverse organs. It exists an inverse relationship between particle size and health hazard [18,19]. The U.S. Environmental Protection Agency (EPA) distinguishes between coarse inhalable particles ≤ 10 µm (PM_10_) and fine inhalable particles ≤ 2.5 µm (PM_2.5_). Moreover, the fraction of particles ≤ 1 µm is defined PM_1_ [20]. 

In previous studies different PM levels within different brands and types of cigarettes were detected [21,22,23]. The strength of tobacco products, the content of tar, nicotine, and carbon monoxide and different additives like aromatics and humectant agents might influence the amount of PM [24]. Based on these findings a comparison of different types of cigarettes with various strengths and ingredients of one special brand seems to be reasonable and necessary. The focus on one single brand minimize interferences by, e.g., different production processes of different manufacturers.

## 2. Materials and Methods 

### 2.1. Tobacco Products

The particle size fractions of PM_10_, PM_2.5_, and PM_1_ of four cigarette types of the brand Marlboro [25] were analyzed in comparison to the reference cigarette 3R4F developed by the Kentucky Research and Development Center (University of Kentucky, USA) [26]. The four cigarette types of Marlboro were as followed: Marlboro Silver Blue, Marlboro Gold, Marlboro Red, and Marlboro Red without additives. They differ among others in filter length and strength (content of tar, nicotine, and carbon monoxide) shown in Table 1. For more detailed information about the ingredients of the Marlboro brands the reader is referred to the Federal Ministry of Food and Agriculture of Germany (Bundesministerium für Ernährung und Landwirtschaft) [27] and Philip Morris USA [28].

### 2.2. Automatic Environmental Tobacco Smoke Emitter (AETSE)

Each 20 cigarettes of 4 Marlboro cigarette types and 20 reference cigarettes were smoked using an automatic environmental tobacco smoke emitter (AETSE). The measurement of PM_10_, PM_2.5_, and PM_1_ took place in a glass chamber with a volume of 2.88 m^3^ serving as an enclosed interior space. The AETSE, a smoke pump for medical research designed and engineered by Schimpf-Ing. Trondheim, Norway [29], is installed in this chamber and allows generating smoke of tobacco products in a reproducible way without exposing the investigator or test persons. 

### 2.3. Smoking Protocol

A modified smoking protocol was used in accordance to the Tobacco Smoke Particles and Indoor Air Quality (ToPIQ) studies [30,31]. A 200 mL glass syringe moved back and forth via a linear actuator by a stepper motor imitates the smoking process. The glass syringe is connected with the mouthpiece of the tobacco product via a Nylon tube (IMI Norgren, Birmingham, UK). Thereby, mainstream smoke can sucked into the syringe and afterwards pressed back into the chamber. Two valves ensure that on suction the air flows exclusively through the tobacco product and on back flowing the smoke reach directly into the chamber without passing the tobacco product. A microcontroller adjusts the puff volume (40 mL), puff flow rate (13 mL/s), puff frequency (2/min), inter puff interval (24 s), and the amount of 9 puffs. The smoking protocol is subdivided in four different phases and starts with the pre-ignition phase with the blank measurement (5 min). Then the cigarette is lighted and smoked in the combustion phase (4 min 22 s), followed by the extinguishing of the cigarette and the post-combustion phase (5 min). Afterwards the chamber is ventilated for at least 5 min in the suction phase by using an industrial suction device before the next cycle starts. 

### 2.4. Measurement Equipment

Via light scattering the PM concentrations are measured by a Grimm Portable Laser Aerosol Spectrometer (LAS) and Dust Monitor model 1.109 (Grimm Aerosol Technik, Ainring, Germany) [32,33]. The measuring point is located 35 cm beside the tobacco product at the same altitude. The mixture of exhaled mainstream smoke and side-stream smoke of the smoldering tobacco product is sucked in the LAS. To avoid blockage of the laser measuring chamber of the spectrometer by high particle concentrations a dilution of 1:10 with compressed air is necessary. Subsequently, the dilution ratio is considered in the data processing. The Grimm spectrometer detects airborne particles with a size from 0.25 µm to 32 µm in real-time. The LAS displays the measured results as particle count (L^−1^) and detailed dust mass fractions in 31 channels (µg/m^3^). Additionally, the data are displayed as inhalable, thoracic, and alveolic (µg/m^3^) according to European Standard EN 481 [34] and as PM_10_, PM_2.5_, and PM_1_ values (µg/m^3^) according to U.S. EPA [20]. Every six seconds the received data are recorded. 

### 2.5. Data Processing

The collected data of the four Marlboro brands and the reference cigarette were statistically analyzed and compared. Therefore, the area under the concentration-time curve (AUC) and the mean concentration (C_mean_) of 20 cigarettes were calculated for each brand. In order to avoid overestimation of PM due to technical handling, the AUC of five randomly chosen cigarettes per brand were searched for so-called artificial peaks. In this study, the proportion of peaks was defined as acceptable if not exceeding the average AUC plateau by 22%. Peaks higher than 2% to 16% than average AUC plateaus were detected. Hence, all measurements were included in data analysis. All tested cigarette type samples passed the D’Agostino-Pearson Test for Gaussian normality (cut-off *p* = 0.05). Additionally, the data were tested for outliers with the ROUT method (Q = 1%). Here, no outlier was detected. Finally, all investigated tobacco products were tested for differences using one-way ANOVA and Tukey’s multiple comparisons test.

## 3. Results

The PM mean of all measured baseline values (clean air) is 0.6 µg/m^3^. For the reference cigarette the measured C_mean_ increases up to 921 µg/m³ (PM_10_), 918 µg/m³ (PM_2.5_), and 852 µg/m³ (PM_1_). The measured C_mean_ of PM_10_ increases up to 1458 µg/m³ (Marlboro Red without additives) and 668 µg/m³ (Marlboro Silver Blue) and in the case of PM_2.5_, 1452 µg/m³ and 667 µg/m³, respectively. For PM_1_ the values raise up to 1263 µg/m³ (Marlboro Red without additives) and 631 µg/m³ (Marlboro Silver Blue). The results of the AUC-PM values and the C_mean_ values are shown in Table 2. Additionally, Figure 1 shows the AUC PM values of all tested cigarette brands in a direct comparison. The distribution pattern of the PM fractions PM_10–2.5_, PM_2.5–1_, and PM_1_ is shown in Figure 2.

The main part of SHS is composed by PM_1_ fraction with 92.50% (reference cigarette 3), 94.46% (Marlboro Silver Blue), 92.17% (Marlboro Gold), 88.77% (Marlboro Red), and 86.63% (Marlboro Red without additives). The measurements of both Marlboro Red brands (with and without additives) show between 33% and 37% higher PM values (C_mean_ and AUC) than the values of the reference cigarette and 50% to 54% higher PM values than the Marlboro Silver Blue and 32% to 36% higher PM values than the Marlboro Gold, respectively. The PM levels of the Marlboro Gold brand are nearly the same as the values of the reference cigarette. In contrast, the Marlboro Silver Blue, the brand with the lowest tar, nicotine, and carbon monoxide amount in this test field, shows 26% to 27% lower PM values compared to the values of the reference cigarette. Table 3 shows the significance grades of the comparisons of all tested tobacco products. Among the Marlboro brands, both types with higher tobacco strength (Marlboro red with and without additives) show a very high significant in comparison to the types with lower strength (Marlboro gold and Marlboro silver blue). 

The PM data for Marlboro brands indicates that the lower the tar, nicotine, and carbon monoxide amounts the lower are the PM levels. Moreover, in this study the tobacco product with additives shows no significant differences in PM amount to the tobacco product without additives and approximately identical tar, nicotine, and carbon monoxide amount.

## 4. Discussion

Various studies conclude that the PM levels in smoking rooms and households increase in a hazardous way [35,36]. According to the WHO Air quality guidelines the daily average concentration should not exceed 25 µg/m^3^ PM_2.5_ [37]. Depending on the cigarette brand the PM concentrations in an enclosed space of 2.88 m³ (capacity of the measuring cabin) were 27- to 58-fold higher than WHO references and up to 1000-fold higher than the baseline values (smoke free air). This illustrates the massive PM burdens under the study conditions.

The U.S. EPA classifies compact cars with a total passenger and cargo volume of 2.832 m³ to 3.087 m³ [38]. This is a fundamentally important aspect of this study design, because the used measuring cabin has a comparable indoor volume and many people smoke in cars. The passive smoke with the contained particulate matter is not only hazardous for the health of smokers but also of passengers, which are often children. The used smoking regime is similar to conditions in a compact car with closed windows and no ventilation or air conditioning. Sendzik et al. [39] performed a study under five different in vivo conditions, in which the car owners smoked a single cigarette in their cars. The conditions were as followed: Closed windows and engine off, and each a 20-min drive with closed windows, all windows opened, with only driver’s window partially opened and all windows closed but with air conditioning. Their results are similar to our study results depending on the condition and ranged between 223 µg/m³ and more than 3800 µg/m³ PM_2.5_ (that means 9 to 152 fold higher than the WHO references). 

In contrast to the above-mentioned in vivo study, the AETSE used in our study ensured reproducible results without exposure of any test person to the produced smoke and any health risks. It should be pointing out that the AETSE is not able to imitate exactly the human smoking behavior and SHS. The mainstream smoke that the smoker inhales will be humidified in the respiratory tract and due to hygroscopic growth the exhaled smoke particles are nearly 1.5-fold larger than the inhaled particles [40,41]. In addition, a differentiation between inhaled and exhaled mainstream smoke is not possible with the AETSE, but SHS consists only of about 15 % mainstream smoke and about 85 % side-stream smoke [42,43]. Hence, the measured PM emissions of the tobacco products are very similar to SHS, because the AETSE is able to imitate side-stream smoke as realistically as possible. Certainly, the used modified smoking protocol differs from other existing protocols like, e.g., the Standard operating procedure for intense smoking of cigarettes by the WHO [44] or the ISO standard for the machine smoking of cigarettes ISO/TR 17219 [45]. However, it must be mentioned that there is yet no “gold standard” for smoking regimes [46,47,48,49].

The aim of this study was to investigate the influence of cigarette strength and additives on PM amount in SHS. To avoid other influences like, e.g., different production processes of various manufacturers on PM emissions as far as possible, it seemed useful to investigate PM of different cigarette types of one brand, in this case the brand Marlboro. All tested Marlboro cigarettes had the same total length and diameter and the same filter length. The Marlboro cigarette type with the lowest tar, nicotine, and carbon monoxide amounts (Silver Blue) showed the lowest measured PM values. The Marlboro brand with the second lowest tar, nicotine, and carbon monoxide amounts (Gold) showed the second lowest measured PM values. The highest PM amounts showed with very similar measured values the two Marlboro Red types (with and without additives), that had the same tar and carbon monoxide amounts and similar nicotine amount. The measured results lead to the assumption that cigarettes with lower strength emit less PM than cigarettes with higher amounts of tar, nicotine, and carbon monoxide. 

The 3R4F reference cigarette had the same total length and diameter as the Marlboro brands, but the filter was with a length of 27 mm six millimeters longer than the filters of the Marlboro brands. Both Marlboro Red types with similar strength as the reference cigarette showed 33% to 37% higher PM values than the reference cigarette. The Marlboro Gold type with a lower strength than the reference cigarette showed PM values similar to the reference cigarette. The filters of all tested cigarette products were cellulose acetate filters without cavity and with triacetin as plasticizer [26,27]. Thus, and given that the filters have been similarly constructed, the PM data lead also to the assumption, that the longer the filter the lower are the PM amounts in SHS. In 2009 Shin et al. found more than 50% lower total particulate matter (TPM) and tar amounts in mainstream smoke of cigarettes with filter compared to cigarettes without filters [50]. TPM means airborne particulate matter with an upper size limit of 100 µm diameter and also includes PM_10_, PM_2.5_, and PM_1_ [51]. Even in 1965 Keith and Derrick showed a reduction of tar and nicotine amounts between 40% and 50% in filter cigarettes in relation to non-filter cigarettes [52]. 

As both Marlboro Red types (with and without additives) with similar tobacco strengths showed very similar PM levels an influence of the additive mixture in Marlboro Red with additives could not be verified. The few studies with respect to effects of additives to PM in tobacco smoke found contradictory results. Some previous studies showed analogical results regarding the influence of additives on PM amount in SHS. Wasel et al. ascertained no significant differences between cigarettes with and without additives [22]. They assumed rather an influence by filter length on PM amounts. Two investigations on cigarettes with and without the additive menthol, Gaworski et al. in 1997 [53] and Gerharz et al. in 2018 [54], found also no significant differences of amounts of PM. In contrast, the results of Rustemeier et al. showed an increase of 13–28% of PM of cigarettes with additives relative to cigarettes without additives [24]. They added 333 commonly used additives to the 1R4F reference cigarette and measured effects of ingredients. An analysis by Wertz et al. in 2011 of previously secret tobacco industry documents revealed that the documents were changed post hoc [55]. The originally statistically findings showed an additive-associated increase of TPM concentrations and toxicity in cigarette smoke with additives. Hence, it seems to be reasonable to investigate in further studies the influence of different additive mixtures in tobacco products on PM emission.

In this study, the major part of measured PM consisted of particles ≤ 1 µm. Keith and Derrick published in 1960 a study with similar results. They found that the most particles in tobacco smoke are sized between 0.1 µm and 1 µm with a peak between 0.2 µm and 0.25 µm [56]. Nazaroff and Klepeis described SHS as mostly 0.02 µm to 2 µm sized particles [57]. Particles of side-stream smoke were characterised with geometric mean diameters of 0.1 µm [58,59]. Manigrasso et al. measured mean particle diameters ranging from 0.1 µm to 0.14 µm in cigarette smoke due to a rapid coagulation of UFPs and phase changes to semi-volatile compounds. They found also that PM_10_ consists mainly of PM_1_ [60]. Haustein and Groneberg described side-stream-smoke with mean diameters of 0.5 µm [61]. It seems that there exists no common agreement on the peak size of tobacco smoke particles. Compared to this study, Protano et al. found very similar PM_1_ mean concentrations of 1544 µg/m^3^ while smoking a single cigarette, but almost no increase of the PM_10–2.5_ and PM_2.5–1_ fractions. They summed up that smoking of even one cigarette lead to very important air pollution also by UFPs [62]. The used LAS Grimm model 1.109 is able to detect particles with a minimum size of 0.25 µm and is common used in monitoring networks and in continuous measurement of PM [63]. This technical limitation resulted in a nonconformity with the definition of the U.S. EPA, where particles down to 0.1 µm are also included. Hence, to detect particles smaller than 0.25 µm a new measurement system would be essential. Subsequent investigations on UFPs in SHS are reasonable, as health effects of UFPs come more and more into focus [64]. 

The used Grimm model 1.109 measures PM including PM_1_ and semi-volatile fractions like, e.g., water, ammonium nitrate, and some organic compounds via light scattering in real time [65]. Because of this ability the LAS allows to detect the PM amount of each single tobacco product. By contrast, the U.S. EPA Federal Reference Methods (FRMs) for detection of PM often use 24 h sample collection followed by gravimetric measurement of collected PM. Another used FRM application is the real time measurement device Tapered Element Oscillating Microbalance (TEOM) Monitor [65,66]. The protocols for measuring PM_10_ and PM_2.5_ in agreement to the European standard EN 12341 for determination of PM is also a gravimetric method [67]. The listed FRM with the Grimm model EDM 180 is a PM measuring method via light scattering [66]. Several studies confirm that the measurement results of a Grimm model 1.107, 1.108, or in this study the used model 1.109 are very similar to the results of a Grimm model EDM 180 or a TEOM Monitor or gravimetric methods [65,68]. In 2007, Fromme et al. described higher PM measuring results by gravimetric methods than by LAS but with high correlations of the rank order of the measuring values [69]. Provided that the method of measurement during a study will not be changed, the measured values of the used Grimm model 1.109 are valid and reliable. 

## 5. Conclusions

In conclusion, smoking of tobacco products leads to a massive increase of PM in enclosed spaces. This study showed also that the higher the amounts of tar, nicotine, and carbon monoxide and probably the shorter the filters, the higher are the levels of PM in SHS. An influence of the additive mixture in the investigated Marlboro cigarette types could not be ascertained. It seems to be reasonable to verify the correlations of ingredients and filter length of tobacco products and the resulting PM in SHS.

## Figures and Tables

**Figure 1 ijerph-16-00263-f001:**
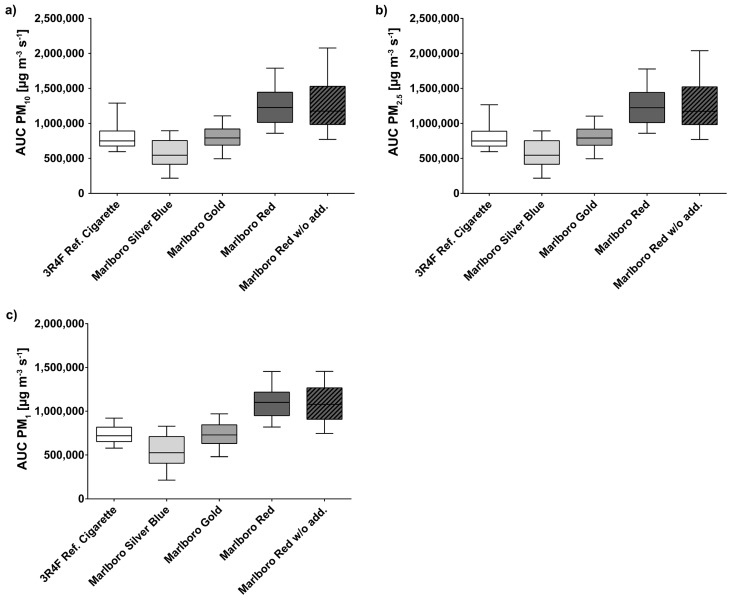
Comparative boxplot (min to max whiskers) of area under concentration–time curve (AUC)–PM of all tested cigarette brands. (**a**) AUC-PM_10_, (**b**) AUC-PM_2.5_, (**c**) AUC-PM_1_.

**Figure 2 ijerph-16-00263-f002:**
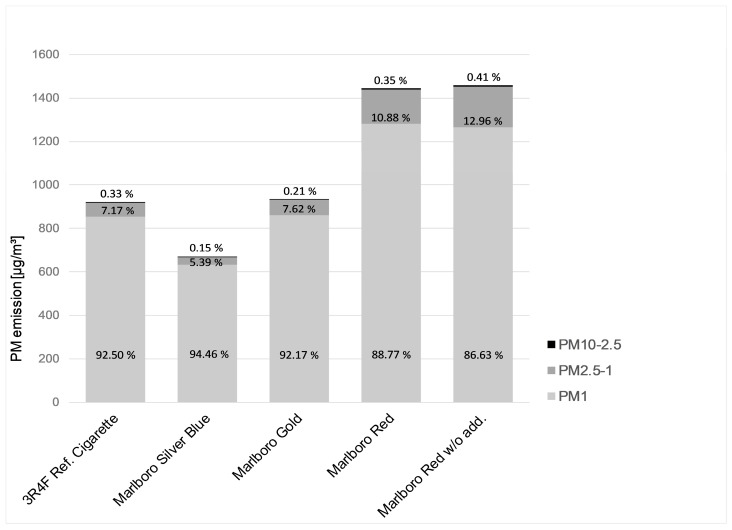
Distribution pattern of PM_10–2.5_, PM_2.5–1_, and PM_1_ of all investigated cigarettes.

**Table 1 ijerph-16-00263-t001:** Characteristics of the investigated cigarette types: The amounts of tar, nicotine, carbon monoxide, the presence of additives, and the dimensions of filter and cigarette are shown.

Ingredients & Dimensions	3R4F Reference Cigarette	Marlboro Silver Blue	Marlboro Gold	Marlboro Red	Marlboro Red without Additives
**Tar (mg)**	9.4	4	6	10	10
**Nicotine (mg)**	0.73	0.4	0.5	0.8	0.9
**Carbon monoxide (mg)**	12	5	7	10	10
**Additives**	yes	yes	yes	yes	no
**Filter length (mm)**	27	21	21	21	21
**Filter diameter (mm)**	8	8	8	8	8
**Cigarette length (mm)**	84	84	84	84	84

**Table 2 ijerph-16-00263-t002:** Area under concentration–time curve (AUC PM_10_, PM_2.5_ and PM_1_) and mean concentrations (C_mean_ PM_10_, PM_2.5_ and PM_1_) with standard deviation of all tested tobacco products.

	3R4F Reference Cigarette	Marlboro Silver Blue	Marlboro Gold	Marlboro Red	Marlboro Red w/o add.
**AUC PM_10_ (µg·m^−3^·s^−1^)**	792,720 ± 152,480	578,280 ± 193,768	806,440 ± 157,991	1234,440 ± 258,690	1,256,570 ± 342,629
**AUC PM_2.5_ (µg·m^−3^·s^−1^)**	790,730 ± 148,547	577,440 ± 193,224	804,940 ± 157,426	1,230,380 ± 255,426	1,251,390 ± 335,957
**AUC PM_1_ (µg·m^−3^·s^−1^)**	733,960 ± 94,781	546,230 ± 169,955	742,580 ± 135,493	1,093,950 ± 173,391	1,088,220 ± 202,603
**C_mean_ PM_10_ (µg·m^−3^)**	921 ± 176	668 ± 223	932 ± 183	1443 ± 307	1458 ± 397
**C_mean_ PM_2.5_ (µg·m^−3^)**	918 ± 172	667 ± 223	930 ± 182	1438 ± 303	1452 ± 389
**C_mean_ PM_1_ (µg·m^−3^)**	852 ± 109	631 ± 196	859 ± 157	1281 ± 215	1263 ± 238

**Table 3 ijerph-16-00263-t003:** Significance level of statistical Tukey’s multiple comparisons test of AUC (PM_10_, PM_2.5_ and PM_1_) for the tested cigarette brands (ns = non-significant, * = *p* < 0.05, ** = *p* < 0.01, **** = *p* < 0.0001).

Paired Comparisons of Tobacco Products	AUC PM_10_	AUC PM_2.5_	AUC PM_1_
**3R4F vs. Marlboro red w/o add.**	****	****	****
**3R4F vs. Marlboro red**	****	****	****
**3R4F vs. Marlboro gold**	ns	ns	ns
**3R4F vs. Marlboro silver blue**	*	*	**
**Marlboro red vs. Marlboro red w/o additives**	ns	ns	ns
**Marlboro red vs. Marlboro gold**	****	****	****
**Marlboro red vs. Marlboro silver blue**	****	****	****
**Marlboro gold vs. Marlboro red w/o add.**	****	****	****
**Marlboro gold vs. Marlboro silver blue**	*	*	**
**Marlboro silver blue vs. Marlboro red w/o additives**	****	****	****

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
