# Peer review of "Particulate Matter Emissions of Four Different Cigarette Types of One Popular Brand: Influence of Tobacco Strength and Additives"

_ijerph, 2019, doi:10.3390/ijerph16020263_

Round 1

Reviewer 1 Report

The topic is relevant but different studies are present in literature on this issue (e.g., see the SI in this journal on tobacco emissions, ed. M. Vitali). The determinations are well-done but the results and relative interpretations are too easy, too scholar. The problems related to PM emissions is largely present in literature, Manigrasso and coauthors proposed different comparisons about it, starting from traditional cigarettes, hand-rolled up to cigars and e-cig. Please, read carefully such papers and make comparisons among those results and results obtained in this paper. The introduction should be improved considering the literature present in such field. Further, the authors should introduce the problems related to submicron particles which are the real problems in this case. Before this paper could be considered for publication, it needs an important improvement according the suggestions above reported.

Author Response

Response to Reviewer 1 Comments

Reviewer 1:

The topic is relevant but different studies are present in literature on this issue (e.g., see the SI in this journal on tobacco emissions, ed. M. Vitali). The determinations are well done but the results and relative interpretations are too easy, too scholar. The problems related to PM emissions is largely present in literature, Manigrasso and coauthors proposed different comparisons about it, starting from traditional cigarettes, hand-rolled up to cigars and e-cig. Please, read carefully such papers and make comparisons among those results and results obtained in this paper. The introduction should be improved considering the literature present in such field. Further, the authors should introduce the problems related to submicron particles which are the real problems in this case. Before this paper could be considered for publication, it needs an important improvement according the suggestions above reported.

Response:

Dear Reviewer 1,

Thank you for the principally positive assessment of our manuscript and the very important hints. We added some mentioned points in the introduction and discussion sections accordingly. Additionally, we added more information in the text of the result section. All changes in the manuscript are highlighted in blue colored text.

Reviewer 2 Report

The manuscript is well written and the study design is well described. However, it reports little relevance as it relates only to 4 types of cigarettes of the same brand. Data obtained and conclusions should be supported by a stat analysis based on a much greater number of cigarette, evaluated by the same methodology.

I suggest reject and resubmit after new experiments

Author Response

Response to Reviewer 2 Comments

Reviewer 2:

The manuscript is well written and the study design is well described. However, it reports little relevance as it relates only to 4 types of cigarettes of the same brand. Data obtained and conclusions should be supported by a stat analysis based on a much greater number of cigarette, evaluated by the same methodology. I suggest reject and resubmit after new experiments.

Response:

Dear Reviewer 2,

Thank you for the positive assessment of the description and the quality of our manuscript as well as the proposal you made. Please, take in mind that we focused on cigarette types of one brand with intent. We wanted to reduce interference factors (e.g. different production processes) as much as possible. Additionally, the investigation of the content of one cigarette pack (n=19 or n=20) in all ToPIQ studies is a consciously compromise between robust statistic and reasonable duration of the experiments. This give us the ability to minimize fluctuating environmental impacts (air humidity, temperature, barometric pressure) by measuring during broadly similar climatic conditions.

All changes in the manuscript are highlighted in blue colored text.

Reviewer 3 Report

This seems to be a good quality paper. My ethical concern with it is that I would prefer the study without names of cigarette brands and concrete products! Having the names there it looks like an hidden advertisement and raises questions toward "no conflict of interest" declaration by authors.  It definitivley allows an interpretation that some cigarettes, concrete names are presented, are less harmful as others. All are harmful....

Author Response

Response to Reviewer 3 Comments

Reviewer 3:

This seems to be a good quality paper. My ethical concern with it is that I would prefer the study without names of cigarette brands and concrete products! Having the names there it looks like an hidden advertisement and raises questions toward "no conflict of interest" declaration by authors. It definitively allows an interpretation that some cigarettes, concrete names are presented, are less harmful as others. All are harmful....

Response:

Dear Reviewer 3,

Thank you for the very positive assessment and the hint not to name the cigarette brand. We changed the title of the manuscript, accordingly:

Particulate matter emissions of four different cigarette types of one popular brand: Influence of tobacco strength and additives

Please, take in mind that the aim of all ToPIQ studies is to investigate the PM emissions of different tobacco products and brands, with or without additives, with different strength, filter length etc. We want to evaluate possible causes that increases PM emissions. We want to name the brand in the text of the manuscript to show which tobacco brand emit more PM.

Regarding the PM burden this study shows that it is better to smoke cigarettes with low strength. You are absolutely right: All tobacco products are harmful. The best way is not to smoke.

All changes in the manuscript are highlighted in blue colored text.

Round 2

Reviewer 2 Report

I read carefully Authors' reply; I keep my previous evaluation:

The manuscript is well written and the study design is well described. However, it reports little relevance as it relates only to 4 types of cigarettes of the same brand. Data obtained and conclusions should be supported by a stat analysis based on a much greater number of cigarette, evaluated by the same methodology. I suggest reject and resubmit after new experiments.

Author Response

Dear Reviewer 2,

Thank you for your second review. We can understand your point of view. On the other hand we have decided consciously to investigate only one package of each cigarette type. We think this is really a good compromise between a robust statistic and a short as possible duration of the investigation to minimize various environmental impacts. This approach have proved itself in practice in all ToPIQ-studies.

All changes in the manuscript are highlighted in blue colored text.